# Synthesis and Characterization of Dental Nanocomposite Resins Filled with Different Clay Nanoparticles

**DOI:** 10.3390/polym11040730

**Published:** 2019-04-22

**Authors:** Alexandros K. Nikolaidis, Elisabeth A. Koulaouzidou, Christos Gogos, Dimitris S. Achilias

**Affiliations:** 1Division of Dental Tissues’ Pathology and Therapeutics (Basic Dental Sciences, Endodontology and Operative Dentistry), School of Dentistry, Aristotle University Thessaloniki, 541 24 Thessaloniki, Greece; koulaouz@dent.auth.gr (E.A.K.); gogos@dent.auth.gr (C.G.); 2Laboratory of Polymer and Color Chemistry and Technology, Department of Chemistry, Aristotle University Thessaloniki, 541 24 Thessaloniki, Greece; axilias@chem.auth.gr

**Keywords:** dental resins, nanocomposite materials, organically modified clays, montmorillonite, intercalation, nanotechnology

## Abstract

Nanotechnology comprises a promising approach towards the update of dental materials.The present study focuses on the reinforcement ofdental nanocomposite resins with diverse organomodified montmorillonite (OMMT) nanofillers. The aim is to investigate whether the presence of functional groups in the chemical structure of the nanoclay organic modifier may virtually influence the physicochemical and/or the mechanical attitude of the dental resin nanocomposites. The structure and morphology of the prepared materials were investigated by means of wide angle X-ray diffraction and scanning electron microscopy analysis. Fourier transform infrared spectroscopy was used to determine the variation of the degree of conversion over time. Measurements of polymerization shrinkage and mechanical properties were conducted with a linear variable displacement transducer apparatus and a dynamometer, respectively. All the obtained nanocomposites revealed intercalated structures and most of them had an extensive filler distribution into the polymer matrix. Polymerization kinetics werefound to be influenced by the variance of the clay organomodifier, whilenanoclays with vinyl groups considerably increased the degree of conversion. Polymerization shrinkage was almost limited up to 50% by incorporating nanoclays. The absence of reactive groups in the OMMT structure may retain setting contraction atlow levels. An enhancement of the flexural modulus was observed, mainly by using clay nanoparticles decorated with methacrylated groups, along with a decrease in the flexural strength at a high filler loading. The overall best performance was found for the nanocomposites with OMMTs containing double bonds. The significance of the current work relies on providing novel information about chemical interactions phenomena between nanofillers and the organic matrix towards the improvement of dental restorative materials.

## 1. Introduction

Composite restorative materials were initially developed to overcome the drawbacks of silicate cements and unfilled resins based on methyl methacrylate monomer and its polymer [1]. Furthermore, aesthetic reasons and concerns associated with amalgam’s toxicity [2] established them as modern biomaterials in the dental industry. These materials consist of the following major components: (a) an organic resin matrix, usually containing 2,2-Bis[p-(2′-hydroxy-3′-methacryloxypropoxy)phenylene]propane (Bis-GMA) or 1,6-bis(methacryloxy-2-ethoxycarbonyl-amino)-2,4,4-trimethylhexane (UDMA) and the co-monomer triethylene glycol dimethacrylate (TEGDMA) as viscosity controller, (b) an inorganic reinforcing filler, such as glass, quartz or fused silica, and (c) a coupling agent, such asγ-methacryloxypropyltrimethoxysilane, to enhance bonding between the filler and resin matrix. The latter contains an activator/initiator system to promote light-activated polymerization of the organic matrix and form cross-linked polymer networks [3,4,5]. In clinical practice, restorations based on dental composite resins are usually challenged by requirements, such as excellent mechanical properties, low wear, and water solubility or sorption, low polymerization shrinkage and marginal leakage, good biocompatibility, caries-inhibition ability and low toxicity, color matching, and stability, etc. [6,7].

Nanotechnology constitutes a promising approach towards the improvement of biomedical [8,9] and dental applications [10,11], utilizing particles as fillers in nanometer scale with a high surface area that can markedly change resins’ macroscopic properties. In recent years, polymer-clay nanocomposites have attracted the strong interest of many materials researchers, as it is possible to achieve impressive enhancements of nanocomposite properties compared to the pure polymers [12,13,14,15,16,17,18]. Particularly, when these properties depend on the surface area of the filler particles, only small amounts (typical less than 5 wt %) of nanoclay may improve mechanical properties and thermal stability, give better resistant to solvents, and decrease gas and liquid permeability [3,19,20,21,22]. Montmorillonite (MMT) is a 2:1 layered silicate, commonly used in polymer nanocomposite formulations. Due to its hydrophilic nature, pristine MMT containing Na^+^ or Ca^2+^ ions is usually modified with quaternary ammonium ions through an ion exchange reaction, and the resulting organomodifiednanoclay (OMMT) is then compatible to the polymer matrix [23,24,25,26,27,28,29,30,31,32]. In general, four types of polymer-clay nanocomposite structures are mainly produced: (1) Exfoliated nanocomposites where the individual nanoclay layers are absolutely delaminated and dispersed in the polymer matrix, while their ordered structure collapses; (2) intercalated nanocomposites formed by the insertion of polymer chains between the intact silicate layers, retaining their regular alternation of galleries and laminas; (3) intermediate nanocomposites which are partially intercalated and partially exfoliated, and (4) conventional composites where layered silicate acts as a conventional micro-sized filler due to the presence of tactoids [3,14,27,33]. Several studies have proven that compared to the intercalated nanocomposites, the exfoliated analogs have a higher Young’s modulus, larger increase in elongation at break, and better thermal stability, and the extent of exfoliation strongly affects the improvement of the final properties [3,34,35,36].

Green composites for environmental purposes [37], drug delivery systems [38], and DNA acid nucleic bases adsorption studies [39] constitute some of the numerous modern MMT applications. There are many reports associated with the incorporation of OMMTs in dental composites and their efficacy in terms of the nanocomposite morphology and final properties. It has been shown that at low nanoclay regimes (Cloisite 93A and 30B up to 10 wt %), the polymerization features, mechanical and thermal properties of the final nanocomposites mainly depend on the degree of exfoliation or intercalation of the clay layers [40]. Light-cured methacrylated/MMT nanocomposites with intercalated or exfoliated structures have also been studied by using the commercial Claytone APA. It was found that materials containing 3 wt % OMMT were extensively cured with increased water uptake, while the presence of the clay had no significant effect on the mechanical properties of nanocomposites [41]. Furthermore, storage modulus and thermal stability increment accompanied with slower polymerization rates and lower degrees of conversion were observed by incorporating up to 15 wt % OMMT with hexyltrimethylammonium bromide [42]. Intercalated nanocomposites containing 50, 60, and 70 wt % Cloisite 10A with lower polymerization shrinkage, high degree of conversion, and lower flexural strength compared to composites analogs filled with silanized silica were also reported [43]. The usage of 50, 60, and 65 wt % Cloisite 30B was found to induce lower polymerization shrinkage than silanized silica [44], while a similar degree of conversion and higher elastic modulus values in comparison to barium glass fillers were indicated at lower concentrations [45]. Dental nanocomposites exhibited high thermal stability when pre-irradiated Cloisite 20A (50 wt %) was used [46]. Moreover, MMT was successfully functionalized with 2-(methacryloyloxy)ethyltrimethylammonium chloride and further incorporated into experimental dental composites. Physical and biological properties results showed a potential interest inthe application of such nanoclays into dental resin composites [47]. The two commercial OMMTs Viscogel B8 and Dellite 67G were better dispersed into dental resin matrix at 2.5 wt % improving the final mechanical properties, whereas the increase of filler concentration reduced the crosslinking ability of the system [48].

In the present study, an effort was made to synthesize and characterize dental nanocomposite resins containing OMMTs with a noticeable diversity of the organic clay modification. For this purpose, a commercially available OMMT with ammonium intercalant having hydroxy-polar groups, an OMMT with C16 alkyl chain hydrophobic ammonium intercalating agent, and two OMMTs containing quaternary ammonium functional methacrylates with C8 and C18 alkyl chains, respectively, were incorporated into dental resin formulations. Moreover, the corresponding surface modified OMMTs with silane coupling agent were also utilized. This work aims to investigate whether the presence of functional groups in the chemical structure of the nanoclay organic modifier may virtually influence the physicochemical and mechanical attitude of dental resin nanocomposites.In a few words, it can be concluded that methacrylated OMMTs are generally responsible for the overall improvement of the produced nanocomposites. The obtained findings could play a key role inthe designing of novel dental nanocomposites suitable to meet the requirements of clinical practice.

## 2. Materials and Methods

### 2.1. Materials

The monomers triethylene glycol dimethacrylate (TEGDMA), 95%, and 2,2-Bis[p-(2′-hydroxy-3′-methacryloxypropoxy)phenylene]propane (Bis-GMA) were both provided by Aldrich, Taufkirchen, Germany. Co-initiator 2-(Dimethylamino)ethylmethacrylate (DMAEMA), 99%, and initiator camphorquinone, 98%, were purchased from J&K Scientific GmbH, (Lommel, Belgium). Commercially available OMMT, Nanomer^®^ I.34MN, produced by Nanocor Company (Hoffman Estates, IL, USA) and supplied by Aldrich (Taufkirchen, Germany), is an –onium ion modified clay containing 25 to 30 wt % methyl dixydroxyethyl hydrogenated tallow ammonium ion. OMMTs with different intercalating organomodifiers, such as cetyltrimethylammonium chloride (MMT-CTAC), dimethylaminooctadecyl methacrylate (MMT-DMAODM), dimethylaminohexadecyl methacrylate (MMT-DMAHDM), as well as two surface modified analogs, with 3-(trimethoxysilyl)propyl methacrylate, S.MMT-CTAC, and S.MMT-DMAHDM were all prepared in our previous works [49,50]. The specific chemical structures of all MMT organomofidiers are represented in Figure 1. All other chemicals used were of reagent grade.

### 2.2. Sample Preparation

Six groups of experimental composites were prepared by initially mixing a Bis-GMA/TEGDMA base (50:50 wt/wt) which containedcamphorquinone (0.2 wt %) and DMAEMA (0.8 wt %) as a photo-initiating system. Afterward, the different organomodified clays were mixed with the resin by manual mixing until the powder was completely wetted with organic matrix, and the obtained mixture was ultrasonicated for 10 min. The nanofiller loading was 50 wt % to ensure paste handling properties almost similar to a commercial dental composite resin. Bis-GMA/TEGDMA pure matrix was also prepared to the same composition to be used as control material. 

### 2.3. Measurements

X-ray diffraction (XRD) analysis of cured materials and nanopowders (MMT-CTAC and S.MMT-CTAC) were performed over the 2*θ* range of 2° to 10°, at steps of 0.05°, and counting time of 5 s per step, using a Miniflex II XRD system from Rigaku Co. (Tokyo, Japan) with Cu Ka radiation (*λ* = 0.154 nm).

Scanning electron microscopy (SEM) was carried out using a JEOL JSM-6390LV (JEOL USA, Inc., Peabody, MA, USA) scanning microscope equipped with an energy-dispersive X-ray (EDX) INCA PentaFETx3 (Oxford Instruments, Abingdon, England) microanalytical system. All the studied surfaces were coated with carbon black to avoid charging under the electron beam.

Polymerization shrinkage kinetics were conducted according to the "bonded-disk" method which was initially published and further refined by Watts and co-workers [51,52,53]. Briefly, a disk-shaped un-set specimen with dimensions of 1.0 mm × 8.0 mm was formed and centrally positioned upon a 3 mm thick rigid glass plate. A flexible cover-slip diaphragm, supported by an outer peripheral brass ring with internal diameter circa 15 mm, was rested on the upper surface of the specimen disk so as to be adherent. A uniaxial LVDT (linear variable displacement transducer) measuring system was positioned centrally onto the cover slip. The signal from the LVDT was transmitted to a computer by a transducer indicator (E 309, RDP Electronics Ltd., Wolverhampton, UK), and a high-resolution analog to digital converter (ADAM-4016 acquisition module) supported by datalogger software (Advantech Adam/Apax.NET Utility, version 2.05.11). Measurements records were taken by continuous irradiation of specimens with a quartz–tungsten–halogen lamp (Astralis 3, Ivoclar-VivadentSchaan, Liechtenstein) at 650 mW·cm^−2^ for 5 min directly from beneath the glass plate at room temperature. A radiometer (Hilux, Benlioglu Dental Inc., Ankara, Turkey) was used to verify the output irradiance of the light-curing device. Four repetitions (n = 4) were made at each specimen. Strain was calculated as:(1)$$\varepsilon \left(\%\right)=100\;\times \frac{\Delta L}{L_{0}}$$
where ε(%) represents the strain (%), ΔL and L_0_ are the shrinkage displacement and the initial specimen thickness, respectively.

Polymerization kinetics wereperformed by placing a small amount of each composite between two translucent Mylar strips, which were pressed to produce a very thin film. The films of unpolymerized composites were exposed to visible light as previously described, and immediately scanned by a Spectrum One Perkin–Elmer FTIR spectrometer (PerkinElmer Inc., Waltham, MA, USA) at different curing time intervals (0, 5, 10, 15, 20, 25, 30, 40, 60, 80, 120, 180 s). Spectra were obtained over 4000–600 cm^−1^ region and acquired with a resolution of 4 cm^−1^ and a total of 32 scans per spectrum. The area of aliphatic C=C peak absorption at 1637 cm^−1^, and the aromatic C=C peak absorption at 1580 cm^−1^ were determined, utilizing a base line technique which proved the best fit to the Beer–Lambert law [54]. The aromatic C=C vibration was used as an internal standard. The percent degree of monomer conversion (DC %) of the cured specimen, which expresses the percent amount of double carbon bond reacted at each time period, was determined according to the equation:(2)$$\mathrm{DC}\left(\%\right)=\left[1-\frac{{\left(\frac{A_{1637}}{A_{1580}}\right)}_{\mathrm{polymer}}}{{\left(\frac{A_{1637}}{A_{1580}}\right)}_{\mathrm{monomer}}}\right]\times 100$$

For flexural tests, bar-specimens were prepared by filling a Teflon mold (2 mm × 2 mm × 25 mm) with unpolymerized paste in accordance withISO 4049. The mold surfaces were overlaid with glass slides covered with a Mylar sheet to avoid air entrapping and adhesion of the final set material. The assembly was held together with spring clips and irradiated by overlapping on both sides, as previously described. Each overlap irradiation lasted for 40 s. Five specimen bars (n = 5) were prepared for each nanocomposite. The specimens were stored at 37 ± 1 °C in dark conditions for 24 h immediately after curing. Afterward, they were bent in a three-point transverse testing rig with 20 mm between the two supports (3-point bending). The rig was fitted to a universal testing machine (Testometric AX, M350-10kN, Testometric Co. Ltd., Rochdale, England). All bend tests were carried out at a cross-head speed of 0.5 mm·min^−1^ until fracture occurred. The load and the corresponding deflection were recorded. The flexural modulus (E) in GPa and flexural strength (σ) in MPa were calculated according to the following equations:(3)$$E=\frac{F_{1}l^{3}}{4{\mathrm{bd}}_{1}h^{3}}10^{-3}\;\mathrm{and}\;\sigma =\frac{3{\;F}_{\max}\;l}{2{\mathrm{bh}}^{2}}$$
where: F_1_ is the load recorded in N, F_max_ is the maximum load recorded before fracture in N, l is the span between the supports (20 mm), b is the width of the specimen in mm, h is the height of the specimen in mm, and d_1_ is the deflection (in mm) corresponding to the load F_1_.

### 2.4. Statistical Analysis

The values of the measured mechanical properties represent mean values ± standard deviation of replicates. Kruskal–Wallis statistic test, followed by a Dunn’s test, for multiple comparisons between means to determine significant differences (p < 0.001), for analysis of the experimental results. This was performed separately for both flexural modulus and strength parameters.

## 3. Results and Discussion

### 3.1. Structure and Morphology Characterization

Diffractogramsfor the synthesized dental nanocomposite resins and their corresponding pure nanoclays appear in Figure 2. The presence of two distinct diffraction peaks is observed for most nanocomposites. Furthermore, the diffraction peaks of the pristine OMMTs are mainly shifted to lower 2*θ* angles (first peaks), denoting a possible obtained intercalated structure due to the insertion of macromolecules within clay galleries. Jlassi et al. have proven the co-existence of exfoliated with intercalated clay layers for epoxy nanocomposites with 3 wt % bentonite acting as intercalated chain transfer agent [55], as well asthe high degree of exfoliation for epoxy resins containing 0.5 wt % bentonite/4-diphenylamine diazonium/polyaniline nanocomposite filler [56]. However, our findings were also enhanced by predominantly intercalation phenomena previously reported by other researchers regarding dental composites filled with MMT nanoparticles at higher clay loadings around 50 wt % [45,46,57]. The secondary peaks of nanocomposites remained in the same angle regions as the initial peaks of the pure OMMTs, implying the portion of nanoclay that did not intercalated by macromolecular chains. Hence, agglomerated MMT nanofillers might be formed usually called as "tactoids". In such relatively high nanofiller loadings, similar structural characteristics were also reported by other researchers [43,44,45].

The *d*_001_-basal spacing values for all nanocomposite resin groups were calculated according to the Bragg’s law (n*λ* = 2dsinθ) and are listed in Table 1. The *d*_001_-values for the majority of the used OMMTs were determined in our previous work [50] and are also available in Table 1 for comparative purpose.

Particularly, the incorporation of MMT-DMAODM nanofillers into dental nanocomposites imposed a considerable change of interlamelar spacing, implying a possible structure composed of highlyintercalated regions along with a small portion of agglomerates. The vinyl groups of DMAODM are capable ofstrongly interacting with methacrylated groups of monomers, and, thus, favor the formation of such structures. Similar performance is also observed for surface modified OMMTs, S.MMT-DMAHDM, and S.MMT-CTAC, as silane coupling agent could also chemically interact with monomers’ functional groups (Figure 3). On the other hand, MMT-DMAHDM nanoparticles showed a satisfactory alternation of *d*_001_-basal spacing values, due to functional end groups, denoting anintercalated structure of nanocomposite. However, this nanocomposite structure seems to be enriched with clay agglomerates as the first diffraction peak is shorter than the secondary. In the same manner, Nanomer^®^ I.34MN and MMT-CTAC exhibited a well-established agglomeration of nanofillers despite their remarkable extent of monomer intercalation.

Figure 4 shows SEM microphotographs taken for all the nanocomposite resins’ groups. It can be seen that voids were presented on the observed surfaces, while clay nanofillers (white dots) were extensively distributed into the polymer matrix for the majority of composites, despite the formation of some agglomerates distinguished as larger white dots. These clusters formed by OMMTs’ aggregation could be attributed to some restrictions associated with the widely used manual mixing [58,59,60,61,62]. Previous studies based on this technique have proven that the relatively high clay filler loadings (varying from 16 up to 70 wt %) might result in the occurrence of a great number of tactoids [43,45,46,63]. Nanomer^®^ I.34MN (Figure 4a), MMT-DMAHDM (Figure 4c), and MMT-CTAC (Figure 4e) clays seem to form the most and largest agglomerates. The lower extent of clustering is observed for OMMTs, such as MMT-DMAODM (Figure 4b), S.MMT-DMAHDM (Figure 4d), and S.MMT-CTAC (Figure 4f), with organomodifiers containing methacrylated groups, which maybe due to their chemical affinity with monomers which can promote better dispersion of nanofillers into the organic matrix. The above observations describe a surface nanoparticles’ distribution which is almost in agreement with the aforementioned XRD results.

### 3.2. Polymerization Kinetics

Figure 5a represents the results on the degree of conversion versus time for Bis-GMA/TEGDMA matrix and the nanocomposite resins containing different types of nanoclayorganomodifiers. Typical FTIR absorbance peaks (1635 and 1582 cm^−1^) recorded for Group 1, 2, and 4 nanocomposites, under cured and uncured conditions, which were used to calculate the degree of conversion are given in Figure 5a. The experimental values of the final DC (%) are listed in Table 2. Representative data for some commercial dental resins [64,65] are also given in Table 2.

According to Figure 5a, for pure Bis-GMA/TEGDMA the reaction conversion eventually reaches the relatively low ultimate value of approximately 62%. This performance is attributed to the glass-effect procedure due to the influence of diffusion-controlled phenomena on the propagation reaction and the constrained mobility of monomer molecules to find and react with a macro-radical [66,67,68,69]. Despite available monomers still remaining unreacted, the reaction rate is almost zero. Furthermore, an abrupt increment in the degree of conversion observed at the first 5 min for the majority of the materials, implies the appearance of the well-known auto-acceleration or gel-effect phenomenon due to the effect of diffusion-controlled phenomena on the termination reaction and the reduced mobility of live macro-radicals to find one another and react [66]. As a result, their concentration increases, leading toelevated reaction rates.

In particular, the gel-effect phenomenon seems to be more effective for MMT-DMAODM, MMT-DMAHDM, and S.MMT-DMAHDM, as the presence of OMMT nanofiller might act as a barrier to the diffusion of macroradicals to find each other and terminate. Thus, the local concentration of radicals increases leading toincreased reaction rates and higher conversion values at the gel-effect period. It is obvious that the high reactivity of MMT-DMAODM nanoparticles arisingfrom the presence of vinyl functional groups in a short C8 alkyl chain of the organomodifier can significantly enhance the auto-acceleration effect up to 68% degree of conversion, while the final reaction degree reaches almost 70%. The lower conversion value of 50% observed for MMT-DMAHDM and S.MMT-DMAHDM at the gel-effect period maybe denotes a lower reactivity of vinyl groups in a larger C16 alkyl chain, although the ultimate conversion value is almost comparative to the Bis-GMA/TEGDMA matrix. From the results it is also apparent that for hydrophobic MMT-CTAC and S.MMT-CTAC, as well as for Nanomer^®^ I.34MN containing hydroxy-groups, the reaction rates decrease and the gel-effect phenomenon becomes weaker, as the corresponding degree of conversion values vary from 30% to 37%, maybe due to a possible occurrence of aggregates as shown from XRD spectra and SEM images, resulting in a lower capability of clay to act as radical scavenger. Moreover, the ultimate reaction degrees deteriorate as the movement of the small monomer molecules decreases, caused by a lower intercalation extent of the OMMTs, so that their diffusion becomes restricted and they do not easily react with macroradicals. The aspect that clay aggregates can act as microfillers affecting absorption and scattering of light and, thus, attenuating the light photo-initiation process, has been suggested by other researchers [70]. In particular, hydroxy-groups of Nanomer^®^ I.34MN nanoclay, are not expected to participate in the addition polymerization reaction and, thus, affect the auto-acceleration and glass effect, even if they could interact with ether oxygen atoms of monomers via hydrogen bond formation. An alternative explanation could be that primary ketyl radicals formed from the degradation of the camphorquinone photo-initiator [71] may react with the quaternary ammonium hydroxyls on the MMT surface by abstracting a hydrogen atom. The alkoxy radicals of OMMT may then scavenge a further primary radical, leading to the decrease of the effective number of primary radicals which can find a monomer molecule and initiate polymerization [72].

On the other hand, degree of conversion curves for S.MMT-CTAC and S.MMT-DMAHDM denote that additional double bonds due to the silane coupling agent on the surface of OMMT nanoclay rather favor the occurrence of both gel and glass effect.

A theoretical model to optimize experimental data used for DC (%) calculations was previously suggested by Ilie and Durner [73]. According to that, the increase of DC can be described by the superposition of two exponential functions, and the correlation function is asfollows:(4)$$y=y_{0}+\alpha \;(1-e^{-\mathrm{bx}})+c(1-e^{-\mathrm{dx}})$$
where *y* is the DC (%), *x* is the polymerization time, the term *y*_0_ represents the y-intercept, while parameters *a*, *b*, *c*, and *d* are modulation factors of the exponential function to optimize the double exponential function on the measured curve. Typical measuring points recorded for nanocomposites filled with Nanomer^®^ I.34MN, MMT-DMAHDM, and S.MMT-DMAHDM were plotted again (Figure 6), and a line of best fit was inserted through all points.

The calculated parameters *a*, *b*, *c*, and *d* for each nanocomposite are given in Table 3. Particularly, the R^2^ values denote a satisfactory correlation between measured and theoretical data. For a long polymerization time (x→∞) Equation (4) is converted to:(5)$$y={\;y}_{0}+\alpha +c$$

The theoretical final DC (%) can now be calculated by combining the specific values *y*_0_, *a*, *c* of Table 3 and Equation (5). By comparing the DC (%) values of Table 2 and Table 3, it can be postulated that the theoretical approach is very close to the obtained experimental data.

### 3.3. Polymerization Shrinkage Kinetics

Setting contraction of dental composite resins, widely known as polymerization shrinkage, should be ideally limited as much as possible because this achievement favors marginal adaptation, reduces the possibility of a breakdown of the bond to the tooth tissues and prevents the occurrence of secondary caries [5]. Figure 7a illustrates the polymerization shrinkage strain plots versus time for Bis-GMA/TEGDMA matrix and nanocomposite resins containing several types of OMMT nanoclays. Figure 7b shows the apparatus used for shrinkage measurements, as was discussed in Section 2.3. The total strain (%) values are listed in Table 2, while corresponding values are also included for several commercially available dental resins [64,65]. As it was expected, the setting contraction proceeds faster for pure matrix than nanocomposites, and reaches an ultimate strain value almost 3-fold higher compared to the majority of nanocomposites. Nanofillers do not participate in the polymerization reaction, and concurrently decrease the concentration of reactive methacrylate groups [2]. The ultimate strain values recorded for the synthesized nanocomposite resins vary from 2.5% up to 3.5%. In terms of traditional composite resins, values of around 1.5% to 3.0% volumetric contraction are typical as opposed to 6% for acrylics [2]. According to XRD results, OMMT nanoclays are subjected to swelling as their platelets are spread apart via polymerization process. As a result, this expansion mechanism increases the free volume inside the clay and might allow for a reduction of polymerization shrinkage as well as residual stresses [44]. A similar attitude for Bis-GMA/nanoclay systems hasalso been reported by other researchers [74,75]. Campos et al. found that dental nanocomposites’ shrinkage isreduced by increasing MMT filler content from 20 to 50 wt %, indicating the influence of clay nanoparticles on polymerization shrinkage [57].

Figure 7a reveals that nanocomposites filled with OMMTs containing quaternary ammonium methacrylates, such as MMT-DMAODM and MMT-DMAHDM, exhibit higher setting contraction than the corresponding MMT-CTAC with hydrophobic ammonium intercalant throughout the photo-polymerization process. Mahmoodian et al. suggested that not only the degree of conversion but also the volume shrinkage are dependent on the type of clay organomodifier [40]. The presence of functional groups in clay nanoparticles accounts for the observed tendency, as more available double bonds can take part in an addition polymerization reaction between nanoclayorganomodifiers and monomers, leading to the formation of more covalent bonds with a smaller length. Thus, the free volume of the nanocomposite resin is reduced, and the polymerization shrinkageincreases. Moreover, the silane coupling agent on the surface of S.MMT-CTAC and S.MMT-DMAHDM contributes additional double bonds throughout the setting reaction in a similar way, enhancing somewhat the polymerization shrinkage of the final nanocomposite resin when compared to MMT-CTAC and MMT-DMAHDM. According to literature data, OMMTs containing OH polar groups performed as effective setting contraction controllers [40] even at high mass fractions [44]. However, nanocomposite filled with Nanomer^®^ I.34MN yielded the highest strain curve, although hydroxy-groups of its organomodifier are not expected to participate in the polymerization reaction. As it was indicated from XRD and SEM results, possible agglomeration of clay lamellae presented in the intercalated structure of resin could readily reduce the expansion and free volume of Nanomer^®^ I.34MN clay, augmenting the shrinkage strain of the set nanocomposite.

### 3.4. Mechanical Properties

The flexural properties of the dental nanocomposite resins produced were studied in relation to the effect of the different nanoclayorganomodifier, at a constant concentration of 50 wt %. Their flexural modulus and strength dependence on the various OMMT type are shown in Figure 8a,b, respectively, while the corresponding mean values accompanied by standard deviations are listed in Table 2. Typical data are also included for some commercially available resins based on previous studies [64,65]. Concerning the flexural modulus, significant statistic differences were found to exist between Bis-GMA/TEGDMA and nanocomposites with MMT-DMAHDM, S.MMT-DMAHDM, and S.MMT-CTAC. Force versus displacement representative plots of flexural behavior for Bis-GMA/TEGDMA matrix and nanoclay filled dental resins are given in Figure 8c. It is obvious that all nanocomposite resins exhibited higher flexural values compared to the pristine Bis-GMA/TEGDMA matrix. This specific trend was supposed to be expected, as MMT nanoparticles may give rise to high stiffness and modulus [76], and, thus, exert the high resistance against the plastic deformation, as well as the stretching resistance of the oriented macromolecular networks into the clay galleries [77,78]. In each nanocomposite case, the quaternary ammonium intercalating agent of the organoclay can perform as the ‘bridge’ connecting the OMMT platelets and inserted macromolecules. The ammonium head groups of the intercalant molecules reside at the silicate layer, and the organic ligands stretch around the silicate surface and target towards the polymer chains [25].

In particular, nanocomposites filled with S.MMT-CTAC, MMT-DMAODM, and S.MMT-DMAHDM, yielded the highest stiffness among the rest of the nanocomposites produced, corresponding to a 105% to 127% increment of flexural modulus when compared to virgin Bis-GMA/TEGDMA. It seems that vinyl groups of the specific organomodifier contributed to a noticeable stiffening of nanocomposite resins, through a copolymerization process between MMT quaternary ammonium intercalants and methacrylated monomers. A lower improvement percentage of flexural modulus was shown for nanocomposites with Nanomer^®^ I.34MN (57%), MMT-DMAHDM (50%), and MMT-CTAC (63%).Despite the potential of hydroxy-groups of Nanomer^®^ I.34MN to interact with monomers through hydrogen bonding, and vinyl groups of MMT-DMAHDM to copolymerize with matrix, the relatively smaller number and size of the clay clusters found for the nanocomposites reinforced with MMT-DMAODM, S.MMT-DMAHDM, and S.MMT-CTAC nanoparticles may account for the most intensive stiffening effect rendered by the presence of MMT. It could be stated that the longer chain length (C16) of quaternary ammonium intercalating agent DMAHDM onto MMT sheets might restrict the nanoparticle mobility of the nanoclay through the organic phase of monomers, rather than the MMT-DMAODM intercalant chain (C8), resulting in a probable additional chemical reaction between the separate nanofillers with reactive vinyl groups, and, thus, to a higher extent of clustering. Discacciati et al. also suggested the formation of clay larger agglomerates at high clay concentration, when they used the stereochemically heavy and reactive vinylbenzyltrimethyl ammonium cation as MMT intercalant, due to the strong covalent bonding between clay sheets [63].

Although MMT nanoparticles were capable of increasing the stiffness for the total of the tested nanocomposite resins compared to the pure Bis-GMA/TEGDMA resin, a considerable weakness of flexural strength was observed (Figure 8b). The corresponding strength values are also listed in Table 2. The flexural strength decrease recorded at 50 wt % clay concentration contrasts with epoxy nanocomposites employing other types of cation-exchanged clays, such asbentonite/4-diphenylamine diazonium/polyaniline nanocomposite, where a filler loading up to 0.5 wt % can significantly improve flexural strength [56]. At high nanofiller clay levels as much as 50 wt %, the co-existence of agglomerates along with clay intercalates into the cross-linked matrix, according to the combination of XRD and SEM results may be responsible for the low resistance of nanocomposites against flexural loadings. It is widely known that agglomeration can give rise to stress formation [79], while it moderates the intercalation phenomena which favors the improvement of mechanical properties [80]. As a result, the dispersion of tactoids in the polymer matrix can act as a conventional filler, and the interfacial adhesion between the organoclay and polymer matrix is not strong enough to withstand large deformations (Figure 8c). A similar trend of flexural strength decrease at high nanofiller fractions was also observed by other researchers [45]. In terms of the type of MMT organomodifier, according to Figure 8b and Table 2 values, it can be seen that the enrichment of clay functionalization with as many as vinyl groups, including silane modifier, might yield somewhat better flexural strength of the nanocomposite when compared to the other tested experimental organoclays. In particular, the better dispersion and a possible co-polymerization of MMT-DMAODM nanoparticles with organic matrix could account for a slight capability of the nanocomposite to resist bending loadings.

## 4. Conclusions

Dental nanocomposite resins were successfully synthesized by inserting different OMMT nanoparticles. The specific type of clay organomodifier was found to affect not only their morphological characteristics but also their physicochemical and mechanical properties. The combination of XRD and SEM results confirmed the intercalation of macromolecular chains between clay platelets, while some agglomerates of clay still remained. Vinyl groups of intercalant and/or silane coupling agent promoted the better dispersion of nanofillers into the resin matrix. Monitoring of polymerization kinetics revealed that the affinity of methacrylatednanoclays with monomers can contribute to the acceleration of the photo-polymerization reaction, improving the final degree of conversion. Particularly, the high reactivity of methacrylated MMT-DMAODM enhanced the gel-effect up to a 68% degree of conversion, while the final reaction degree reached almost 70%. Experimental data obtained for nanocomposites with Nanomer^®^ I.34MN, MMT-DMAHDM, and S.MMT-DMAHDM were found to be very close to those derived from theoretical calculations. Polymerization shrinkage was lowered by incorporating any kind of the tested nanoclay, especially in the absence of organomodifier reactive groups. Nanocomposite containing the non-reactive hydrophobic MMT-CTAC exhibited the lowest total strain (2.51%), whereas Nanomer^®^ I.34MN yielded the highest strain curve (3.46%) due to a possible agglomeration of clay lamellae.Regarding the mechanical properties, it was verified that the incorporation of OMMT nanofiller improves the stiffness of the dental composite resin. A 105% to 127% increment of flexural modulus values was achieved for nanocomposites reinforced with thepolymerizable S.MMT-CTAC (3.48 GPa), MMT-DMAODM (3.32 GPa), and S.MMT-DMAHDM (3.14 GPa) nanoclays, when compared to Bis-GMA/TEGDMA resin matrix (1.53 GPa). However, the flexural strength was decreased due to the aggregation of clay nanoparticles at high concentrations. The significance of the current work relies on providing novel information about chemical interactions phenomena between nanofillers and organic matrix towards the improvement of dental restorative materials.

## Figures and Tables

**Figure 1 polymers-11-00730-f001:**
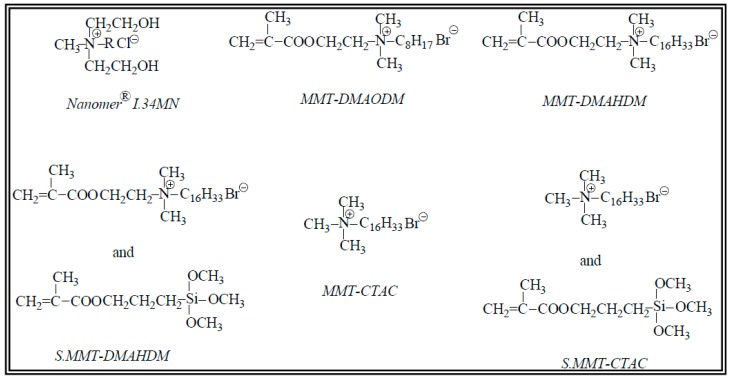
Different types of clay organomodifiers used for nanocomposite synthesis (R stands for hydrogenated tallow).

**Figure 2 polymers-11-00730-f002:**
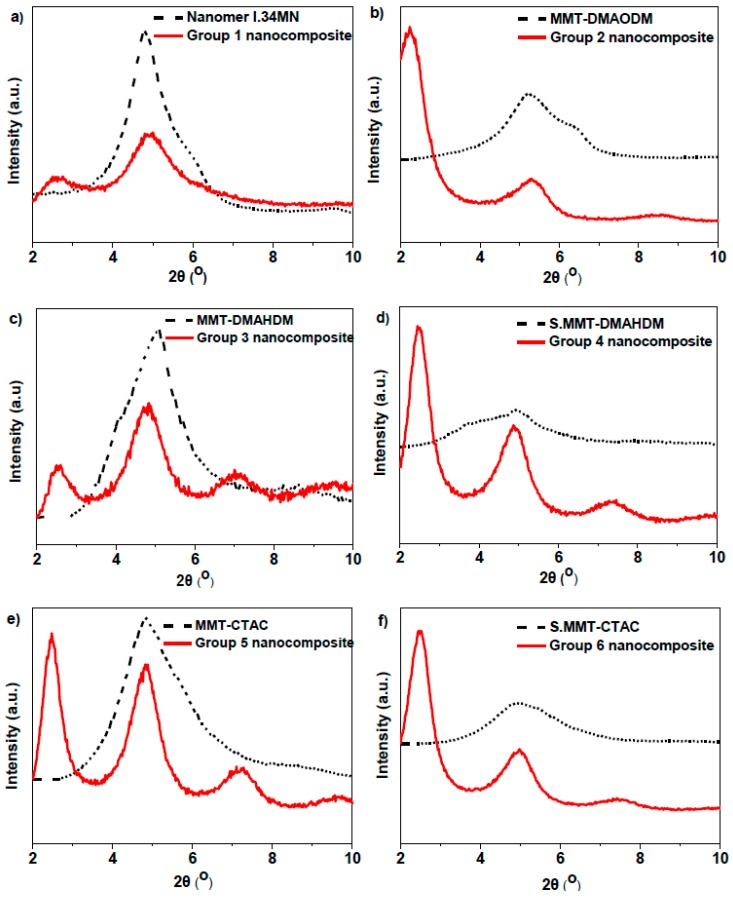
X-ray diffraction (XRD) patterns of the dental nanocomposite resins and the different incorporated organo modified montmorillonite (OMMT) nanoclays.

**Figure 3 polymers-11-00730-f003:**
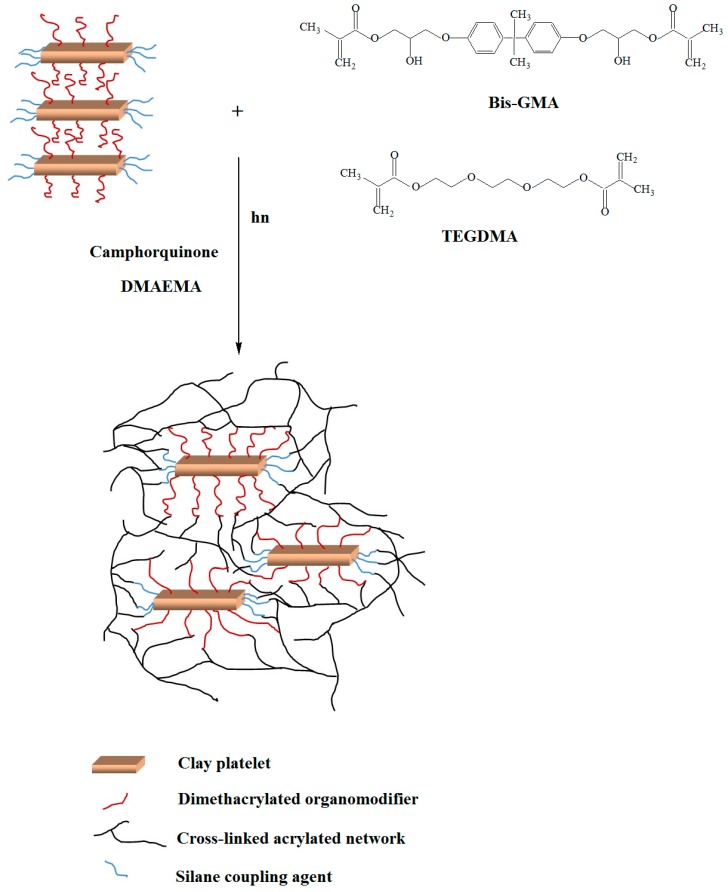
Schematic representation of nanocomposite formation indicating the possible chemical interactions between nanoclayorganomodifier and dental resin monomers.

**Figure 4 polymers-11-00730-f004:**
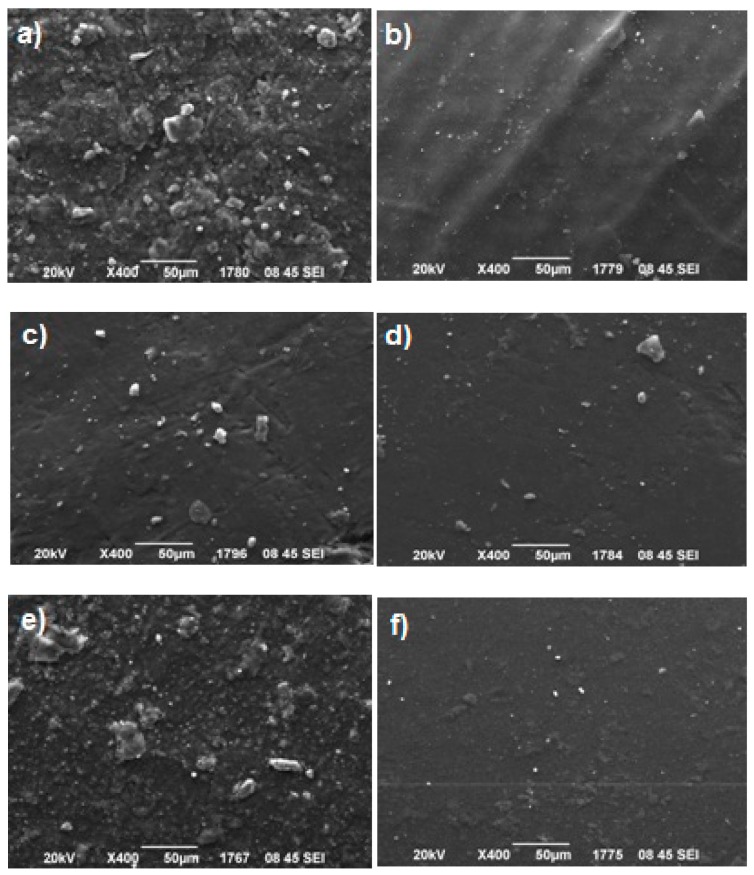
Scanning electron microscopy (SEM) images of a series of experimental dental nanocomposite resins containing (**a**) Nanomer^®^ I.34MN; (**b**) Montmorillonite-dimethylaminooctadecyl methacrylate (MMT-DMAODM); (**c**) Montmorillonite-dimethylaminohexadecyl methacrylate (MMT-DMAHDM); (**d**) S.MMT-DMAHDM; (**e**) Montmorillonite-cetyltrimethylammonium chloride (MMT-CTAC); (**f**) S.MMT-CTAC at concentration 50 wt %.

**Figure 5 polymers-11-00730-f005:**
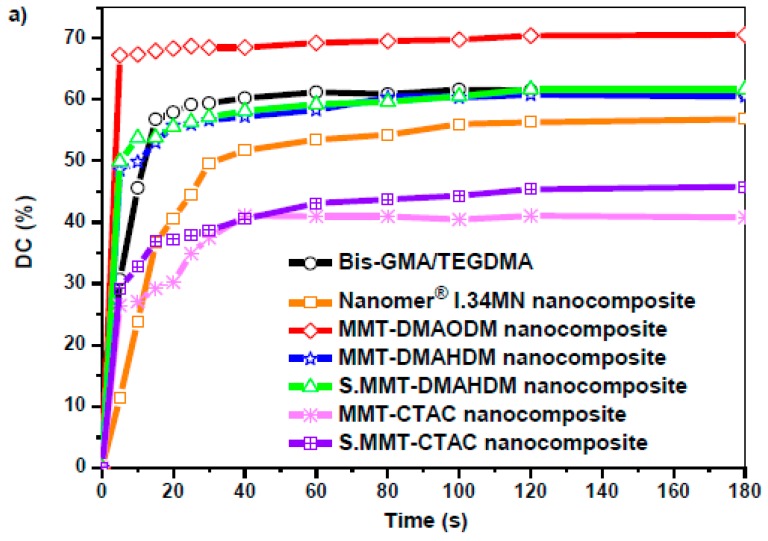
(**a**) Degree of conversion versus time of 2,2-Bis[p-(2′-hydroxy-3′- methacryloxypropoxy)phenylene]propane/triethylene glycol dimethacrylate(Bis-GMA/TEGDMA) matrix and dental nanocomposite resins filled with different OMMT nanoparticles; (**b**) FTIR spectra with measured peak areas (1635 and 1582 cm^−1^) used to calculate the percent degree of conversion (DC (%)) for uncured and cured nanocomposites.

**Figure 6 polymers-11-00730-f006:**
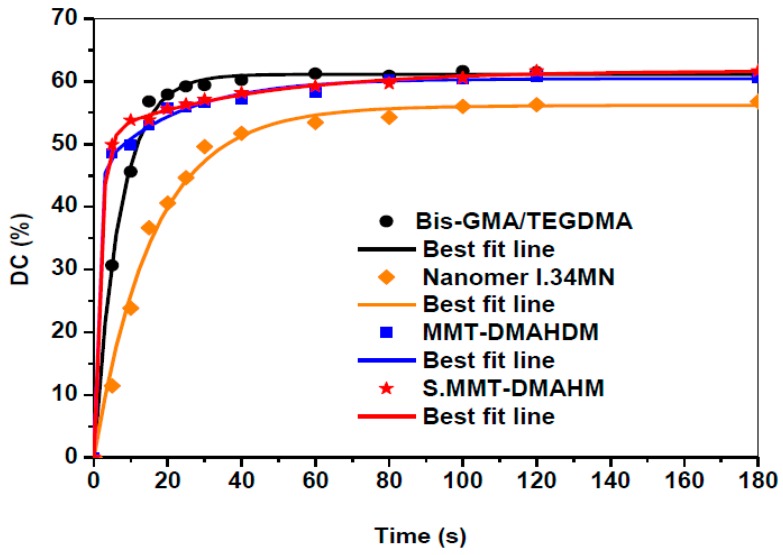
Representative DC-time experimental data and the two exponential approximate functions, for pure Bis-GMA/TEGDMA matrix and nanocomposites filled with Nanomer^®^ I.34MN, MMT-DMAHDM and S.MMT-DMAHDM. Lines of best fit are drawn through all experimental points of the approximate function.

**Figure 7 polymers-11-00730-f007:**
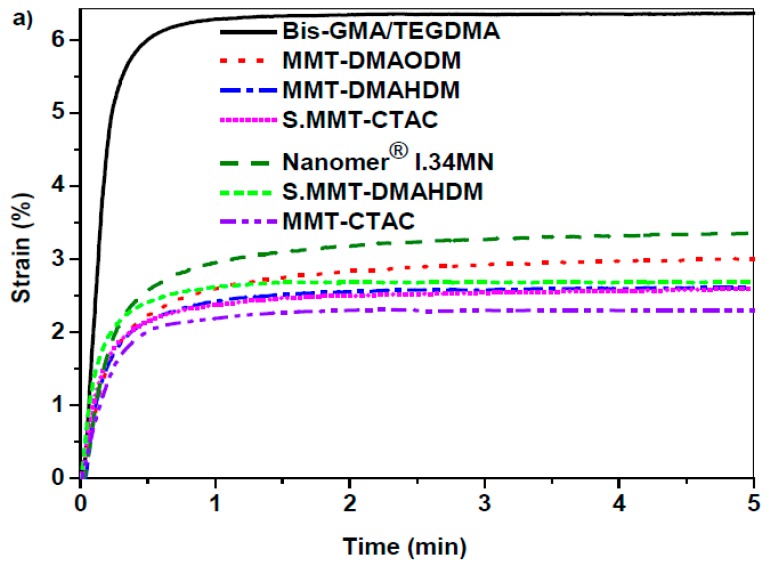
(**a**) Time dependence of polymerization shrinkage strain of Bis-GMA/TEGDMA matrix and dental nanocomposite resins with diverse types of OMMT nanoclays; (**b**) Apparatus utilized for polymerization shrinkage measurements.

**Figure 8 polymers-11-00730-f008:**
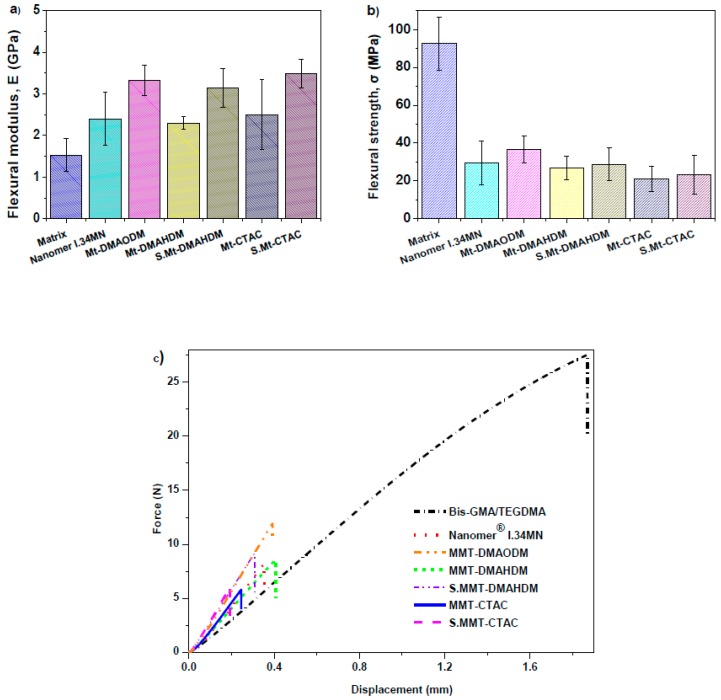
(**a**) Flexural modulus; (**b**) Flexural resistance; (**c**) Graphic representation of force against the displacement of flexural properties, for pure Bis-GMA/TEGMA matrix and dental nanocomposite resins filled with different OMMT nanofillers.

**Table 1 polymers-11-00730-t001:** *d*_001_-spacing values calculated on the basis of X-ray diffraction (XRD) spectra for different species of nanoclays incorporated in nanocomposite resins.

Nanocomposite Resin	OMMT Nanofiller (50 wt %)	*d*_001_ (nm)		Δ*d*_001_ (nm)
		Pure OMMT [49,50]	OMMT into Nanocomposite	
Group 1	Nanomer^®^ I.34MN	1.86	3.39	1.53
Group 2	MMT-DMAODM	1.77	3.95	2.18
Group 3	MMT-DMAHDM	1.73	3.45	1.72
Group 4	S.MMT-DMAHDM	1.78	3.62	1.84
Group 5	MMT-CTAC	1.83	3.56	1.73
Group 6	S.MMT-CTAC	1.79	3.60	1.81

**Table 2 polymers-11-00730-t002:** Experimental values concerning curing kinetics, polymerization shrinkage and mechanical properties of pure 2,2-Bis[p-(2′-hydroxy-3′- methacryloxypropoxy)phenylene]propane/triethylene glycol dimethacrylate(Bis-GMA/TEGDMA) resin and dental nanocomposite resins.For flexural modulus and strength mean values, the groups with the same superscript letters exhibit statistically significant differences (p < 0.001). Literature data for typical commercially available dental resins are also included [64,65].

Sample	Final DC (%)	Total Strain (%)	Flexural Modulus (GPa)	Flexural Strength (MPa)
Bis-GMA/TEGDMA	61.60	6.47 ± 0.19	1.53 ± 0.40 ^a,b,c^	92.59 ± 13.94 ^d,e,f^
Nanomer^®^ I.34MN nanocomposite	56.80	3.46 ± 0.37	2.40 ± 0.64	29.45 ± 11.45
MMT-DMAODM nanocomposite	70.60	2.80 ± 0.23	3.32 ± 0.36 ^a^	36.60 ± 7.28
MMT-DMAHDM nanocomposite	60.80	2.71 ± 0.13	2.30 ± 0.15	26.82 ± 6.33 ^d^
S.MMT-DMAHDM nanocomposite	61.70	2.86 ± 0.37	3.14 ± 0.47 ^b^	32.39 ± 3.52
MMT-CTAC nanocomposite	41.00	2.51 ± 0.29	2.50 ± 0.84	21.14 ± 6.79 ^e^
S.MMT-CTAC nanocomposite	45.80	2.54 ± 0.22	3.48 ± 0.34 ^c^	23.09 ± 10.23 ^f^
Filtek^TM^ Z350 XT (3M ESPE, St. Paul, MN, USA) [64]	50.96	1.66 ± 0.15	9.13 ± 0.66	80.52 ± 15.88
Tetric^®^ N-Ceram Bulk Fill (Ivoclar-Vivadent, Schaan, Liechtenstein) [64]	49.50	1.36 ± 0.08	7.05 ± 0.60	60.37 ± 11.05
Tetric^®^Evo Ceram Bulk Fill (Ivoclar-Vivadent, Schaan, Liechtenstein) [65]	56.70	-	6.10	94.50
Grandio (Voco, Cuxhaven, Germany) [65]	62.80	-	15.30	125.00

**Table 3 polymers-11-00730-t003:** Parmeters describing the exponential sum function accompanied by the coefficient of deterimination, R2, and the calculated values of the final percent degree of conversion (DC (%)).

Resin	*y* _0_	*α*	*b*	*c*	*d*	*R* ^2^	Final DC (%)
Bis-GMA/TEGDMA	−0.2870	6.4892	0.1446	54.9137	0.1447	0.9972	61.69
Nanomer^®^ I.34MN nanocomposite	−1.8527	41.3735	0.0652	16.4640	0.0652	0.9929	55.98
MMT-DMAHDM nanocomposite	0.0000	15.1918	0.0462	45.2200	1.0484	0.9982	60.41
S.MMT-DMAHDM nanocomposite	0.0000	10.2364	0.0243	51.4829	0.5863	0.9995	61.72

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
