# Peer review of "Synthesis and Characterization of Dental Nanocomposite Resins Filled with Different Clay Nanoparticles"

_polymers, 2019, doi:10.3390/polym11040730_

Round 1

Reviewer 1 Report

Dear Editor,

dear Authors,

The manuscript by Nicolaidis et al. reports on dental nanocomposites resins. The work is well done and the obtained results, even if some of them are somehow expected, are suitable for the scientific community.

Many properties were investigated in a proper way and the given explanations on the results are convincing.

The reason for this study is well explained in the introduction part.

It is a pitty that the flexural strenght was decreased due to the clustering of the nanoparticles, but at high concentrations it is normal.

The MS is clear and the subject fits to the journal scope. 

From the scientific point of view I have just one objection. Afterwards, only a formal issue.

Therefore I recommend the MS for publication in Polymers after major revision as listed below.

Just one objection:

Could the authors please add in table 2 more already used (commercial) dental resins. The readers could have a better understanding and could better compare the obtained results. 

Besides:

Just some Typos are detectable:

for instance: page 5 line 224 ...Driffractograms

If the authors want to add relevant references in the introduction part on new applications for MMT,  I provided a short list of new and well cited papers.

1) Ceramics InternationalVolume 45, Issue 2, 1 February 2019, Pages 2751-27592

2) International Journal of PharmaceuticsVolume 457, Issue 1, 2013, Pages 224-236

3) Applied Clay ScienceVolume 53, Issue 4, October 2011, Pages 657-668

All the best,

nice work.

Author Response

Authors are thankful to reviewer about the precious working time spending on the assessment of this study and giving specific advices that contribute to improvement of the paper structure and quality of our research.

Comment 1:Just one objection:

Could the authors please add in table 2 more already used (commercial) dental resins. The readers could have a better understanding and could better compare the obtained results”. 

Response 1: Table 2 (page 10) was enriched with data related to two commercially available dental resins, namely the nanofilled composite FiltekTM Z350 XT (3M ESPE, St. Paul, MN, USA), Tetric® N-Ceram Bulk Fill and Tetric® Evo Ceram Bulk Fill composites (Ivoclar-Vivadent, Schaan, Liechtenstein), and hybrid composite Grandio (Voco, Cuxhaven, Germany), so as to help readers having a better understanding and making relative comparisons.

Comment 2:Besides:

Just some Typos are detectable:

for instance: page 5 line 224 ...Driffractograms”

Response 2: The word “Driffractograms” was corrected to “Diffractograms” (page 5, line 228).

Comment 3: If the authors want to add relevant references in the introduction part on new applications for MMT, I provided a short list of new and well cited papers”.

Response 3: New applications for MMT were added in the introduction part according to reviewer’s suggestions (page 2, lines 76-77).

Reviewer 2 Report

This paper studied dental nanocomposites by both experimental and theoretical works. It is well organized and can be accepted.

The conclusion needs to be extended with more details.

Author Response

Authors actually appreciate the reviewer’s evaluation of the manuscript's quality and checking for possible improvements. For this purpose, we proceeded to the essential changes according to the reviewer's following comment: 

Comment:The conclusion needs to be extended with more details”.

Response: New information based on experimental data were embedded into the conclusion part in order to be extended with more details (page 17, lines 582-585, 587-589, 591-594).

Reviewer 3 Report

This is an interesting paper on clay-nanocomposites for dental materials and could be included in Polymers after revision. One interesting point is the systematic study of polymerization kinetics and degree of conversion.

1) The phrasing should be improved. For example, what is the meaning of :

"On the contrary, flexural modulus augmentation lowered when nanoclays such as Nanomer® I.34MN (57%),...", see lines 519-520. The whole paper could be improved after careful editing.

2) I am surprized that exfoliation has not been achieved for the nanocomposites. This could be due to the use of surfmers instead photoinitiator-modified nanoclays.

I invite the authors to check and cite: European Polymer Journal 72 (2015) 89–101;

Langmuir 2016, 32, 3514−3524;

although these are papers on clay-polymer nanocomposites are not intended but they are relevant in terms of chemistry and mechanical properties.

3) Returning to the use of surfmers, again I can see in figure 8 decrease of lexural strength; this contrasts with nanocomposites employing other types of cation-exchanged clays. See for example Langmuir 2016, 32, 3514−3524;.

4) There are not much papers cited on the subject from the recent years 2017-2019. 

The authors should cite more relevant papers and check relevant papers published by Polymers (MDPI). I have found 31 relevant papers with keyword "dental".

The paper in Journal of Thermal Analysis and Calorimetry 2018, Vol 131, pp 771–774 is relevant and should be discussed.

5) The paper does not tell what is the significance of the work. This should be stated at the end of the Abstract and the end of the Conclusion in order to improve the impact of the paper. 

Author Response

Authors would like to thank the reviewer for the devote time to reading and making comments regarding the manuscript. We strongly believe that the related suggestions will be decisive to the improvement of our research.

Comment 1: “The phrasing should be improved. For example, what is the meaning of :

"On the contrary, flexural modulus augmentation lowered when nanoclays such as Nanomer® I.34MN (57%),...", see lines 519-520. The whole paper could be improved after careful editing”.

Response 1: The phrase: "On the contrary, flexural modulus augmentation lowered when nanoclays such as Nanomer® I.34MN (57%), MMT-DMAHDM (50%), MMT-CTAC (63%) were incorporated into Bis-GMA/TEGDMA matrix."

was replaced by the phrase:

"A lower improvement percentage of flexural modulus was shown for nanocomposites with Nanomer® I.34MN (57%), MMT-DMAHDM (50%), and MMT-CTAC (63%)."

in order to indicate the less intensive stiffening effect rendered by these clays in comparison to MMT-DMAODM, S.MMT-DMAHDM and S.MMT-CTAC nanoparticles (page 16, lines 535-539).

Comment 2:I am surprised that exfoliation has not been achieved for the nanocomposites. This could be due to the use of surfmers instead photoinitiator-modified nanoclays.

I invite the authors to check and cite: European Polymer Journal 72 (2015) 89–101;

Langmuir 2016, 32, 3514−3524;

although these are papers on clay-polymer nanocomposites are not intended but they are relevant in terms of chemistry and mechanical properties”.

Response 2: The relevant papers proposed by reviewer were checked and cited. Evidence was also given for the possible existence of intarcalated instead of exfoliated structures (page 5, lines 232-238).

Comment 3: “Returning to the use of surfmers, again I can see in figure 8 decrease of lexural strength; this contrasts with nanocomposites employing other types of cation-exchanged clays. See for example Langmuir 2016, 32, 3514−3524;”.

Response 3: A comparison was given between the trend of experimental flexural strength and the literature data (Langmuir 2016, 32, 3514−3524) as suggested by the reviewer (page 16, lines 555-558).

Comment 4:There are not much papers cited on the subject from the recent years 2017-2019. 

The authors should cite more relevant papers and check relevant papers published by Polymers (MDPI). I have found 31 relevant papers with keyword "dental".

The paper in Journal of Thermal Analysis and Calorimetry 2018, Vol 131, pp 771–774 is relevant and should be discussed”.

Response 4: The relevant papers were cited in the introduction part (page 2, lines 54-55) from the recent years 2017-2019 published by Polymers (MDPI).

The relevant paper in Journal of Thermal Analysis and Calorimetry 2018, Vol 131, pp 771–774 was also cited and discussed (pages 13-14, lines 453-455 and page 5 lines 236-238).

Comment 5: “The paper does not tell what is the significance of the work. This should be stated at the end of the Abstract and the end of the Conclusion in order to improve the impact of the paper”.

Response 5: The significance of the current research was stated at the end of the abstract and the end of the Conclusion (page 1, lines 32-34 and page 17, lines 595-597).

Round 2

Reviewer 1 Report

Dear Editor,

In the resubmitted manuscript the authors answered point by point to all of the reviewers considerations.

It is evident that the authors spent time to answer in a proper way and to enhance the quality of their work. 

This paper is recommended for publication.

Reviewer 3 Report

The Authors have addressed all concerns of this Reviewer and the revised version can be published in Polymers.

Final editing should be done at the galley proofs correction stage.